# Hydrogeochemical Characteristics and Water Quality Evaluation of Carboniferous Taiyuan Formation Limestone Water in Sulin Mining Area in Northern Anhui, China

**DOI:** 10.3390/ijerph16142512

**Published:** 2019-07-14

**Authors:** Meichen Wang, Herong Gui, Rongjie Hu, Honghai Zhao, Jun Li, Hao Yu, Hongxia Fang

**Affiliations:** 1School of Earth and Environment, Anhui University of Science and Technology, Huainan 232001, China; 2National Engineering Research Center of Coal Mine Water Hazard Controlling (Suzhou University), Suzhou 234000, China; 3Wanbei Coal-Electricity Group Co. Ltd., Suzhou 234000, Anhui, China; 4Anhui Provincial Bureau of Coal Geology Hydrologic Exploration Team, Suzhou 234000, China; 5School of Resources and Environmental Engineering of Hefei University of Technology, Hefei 232000, Anhui, China

**Keywords:** Taiyuan formation limestone water, hydrochemical characteristics, water quality type, fuzzy comprehensive evaluation, Huaibei coalfield

## Abstract

The Taiyuan formation limestone water in the Huaibei coalfield is not only the water source for coal mining, but also the water source for industry and agriculture in mining areas. Its hydrogeochemical characteristics and water quality are generally concerning. In this paper, conventional ion tests were carried out on the Taiyuan formation limestone water of 16 coal mines in the Sunan and Linhuan mining areas of the Huaibei coalfield. Piper trigram, Gibbs diagram and an ion scale coefficient map were used to analyze the hydrogeochemical characteristics of the Taiyuan formation limestone water. The water quality was evaluated in a fuzzy comprehensive manner. The results show that the main cation and anion contents in the Taiyuan formation limestone water were Na^+^ > Mg^2+^ > Ca^2+^ > K^+^, SO_4_^2^^−^ > HCO_3_^−^ > Cl^−^. There were differences in the hydrogeochemical types of the Taiyuan formation limestone water in the two mining areas; HCO_3_-Na type water was dominant in the Sunan mining area and SO_4_·Cl-Na type water was dominant in the Linhuan mining area. The chemical composition of the Taiyuan formation limestone water is mainly affected by the weathering of the rock and is related to the dissolution of the evaporated salt and the weathering of the silicate. The fuzzy comprehensive evaluation results show that the V-type water accounts for a large proportion of the Taiyuan formation limestone water in the study area and the water quality is poor. This study provides a basis for the development and utilization of the Taiyuan formation limestone water and water environmental protection in the future.

## 1. Introduction

At present, coal is the main energy used in China, accounting for about 70% of China’s primary energy consumption [1]. Coal mining affects the runoff conditions of groundwater [2,3], and leads to changes in groundwater quality. For mining areas, groundwater is the main source of domestic water. It is of great significance to make clear the characteristics of groundwater chemical evolution and water quality status caused by coal mining.

In recent years, many scholars have conducted in-depth research on groundwater quality and hydrochemical evolution in mining areas. Liu et al. [4] used hierarchical cluster analysis and principal component analysis combined with the inverse modeling method to find a variety of constraints to control the chemical properties of groundwater. Yang et al. [5] studied groundwater hydrochemistry in the Ordos mining area by Piper trigram, main ion ratio and other methods, and found that during the decade of large-scale coal mining, groundwater type changed from HCO_3_ to SO_4_·Cl. Daniel et al. [6] determined the evolution path of groundwater hydrochemistry by combining the hydrogeochemical theory with inverse geochemical modeling. Lin et al. [7] used the sodium percentage (%Na) and sodium adsorption ratio (SAR) parameters to study the deep groundwater samples collected from the coal-bearing aquifer in the Linhuan mining area, and found that the water had higher salinity, alkalinity and poor water quality, and therefore could not be directly used for irrigation and drinking. Fu et al. [8] used the Gibbs map combined with isotope geochemical analysis to clarify the main controlling factors affecting the hydrogeochemical evolution of groundwater and discussed the main sources of SO_4_^2^^−^ in water. Guan et al. [9] carried out water environmental quality assessments on the water in the goaf of the Datong coalfield, Shanxi Province by using the ion proportional coefficient and the improved fuzzy comprehensive evaluation method. Li et al. [10] conducted an environmental quality evaluation of goaf water in the Datong mining area of Shanxi Province by using a single factor evaluation method and improved Nemero pollution index method, which provided the scientific basis for water resources utilization in the goaf area of the coal mine. Qiu et al. [11] studied trace elements in shallow groundwater in the Linhuan mining area and used the health risk assessment model of the US Environmental Protection Agency to scientifically evaluate the carcinogenic risk of shallow groundwater, providing useful references for rational development, utilization and protection of shallow groundwater resources.

In the Huaibei coalfield of Anhui Province, China, the aquifers affected by coal mining include the Quaternary pore aquifer of the Cenozoic, the Permian coal-series sandstone fissure aquifer and the Carboniferous and the Ordovician limestone karst aquifer [12,13]. So far, a great deal of the literature has focused on hydrogeochemical studies of Quaternary pore aquifers and Permian coal-series sandstone fissure aquifers [7,14]; however, there is limited research on hydrogeochemical evolution and water quality evaluation of limestone karst groundwater in mining areas. This paper conducts research on the hydrogeochemical characteristics, evolution studies and water quality of karst groundwater of the carboniferous Taiyuan formation in two mining areas, Sunan and Linhuan (hereinafter referred to as “Sulin mining area”), of the Huaibei coalfield. Its status is evaluated to provide a basis for the development and utilization of the Taiyuan formation limestone water in the study area and water environmental protection.

## 2. Study Area

The Huaibei coalfield Sulin mining area is located in the northern part of Anhui Province, located between 116°15′ to 117°12′ E, and 33°20′ to 33°42′ N. There are seven coal mines in the Sunan mining area and 12 coal mines in the Lihuan mining area (Figure 1). It belongs to the semi-humid climate in the northern warm belt, with an annual average temperature between 14 and 15 °C; the highest temperature is 42 °C and the lowest temperature is −14 °C. The annual average precipitation is 840 mm, and the annual minimum precipitation is 520 mm, most of which are concentrated in the summer.

The limestone of the Carboniferous Taiyuan Formation in the Sulin mining area of Huaibei coalfield is located in the lower part of the Permian coal-bearing strata. The karst and fissures are developed, water-rich and strong. They are not only the main source of water in the mining area, but also the source threatening coal mining.

## 3. Sampling Test and Research Method

### 3.1. Sampling and Testing

In the summer of 2018, a total of 16 Taiyuan formation limestone water samples were collected from 16 coal mines in the Sulin mining area, including Yuanyi (S1), Qingdong (S2), Tongting (S3), Linhuan (S4), Renlou (S5), Yangliu (S6), Wugou (S7), Jiegou (S8), Suntuan (S9), Yuaner (S10), Qianyingzi (S11), Zouzhuang (S12), Qinan (S13), Taoyuan(S14), Zhuxianzhuang (S15), and Luling (S16).

Before sampling, the sampling bucket was rinsed three times with deionized water, and then rinsed three times with sampled water. After sample collection, the pH and TDS (total dissolved solids) of the water samples were tested on site with portable instruments from OHAUS (Shanghai, China). Portable instruments to test pH and TDS were ST20 and ST20T-B, respectively. The measurement accuracy of ST20 reaches 0.05 pH, and that of ST20T-B reaches 1 mg/L. All samples were sent to the laboratory within 24 h and stored in a refrigerator at 4 °C for further testing.

The contents of Na^+^, K^+^, Ca^2+^, Mg^2+^, Cl^−^, and SO_4_^2^^−^ in the water were tested by an ion chromatograph (ICS—600—900) after filtration by 0.45 micron membrane. The contents of CO_3_^2^^−^ and HCO_3_^−^ were determined by acid-base titration.

### 3.2. Research Methods

There are many indicators involved in water quality assessment, and the weights of each indicator are different; therefore, it is not reasonable to use a single indicator for water quality assessment [15]. In recent years, the fuzzy comprehensive evaluation method has been widely applied to multi-index water quality evaluations [9,16,17]. This method is an evaluation method based on the actual measured data and the limited value in the water quality standard, using the principle of fuzzy transformation and the principle of maximum membership degree to comprehensively process the measured data [18]. However, when a single factor is seriously polluted, the traditional fuzzy comprehensive evaluation results will also be affected [19]. Thus, this paper uses the improved fuzzy comprehensive evaluation method. The specific steps are as follows:

(1) Establish factor subsets and evaluation language sets

Firstly, a set of factors is established according to the measured values of pollutants U = {U_1_, U_2_, …, U_n_}. The evaluation standard is based on the Chinese national groundwater quality standard (GB/T 14848-2017) [20], establishing an evaluation set V = {I, II, III, IV, V}, as shown in Table 1.

(2) Establish a fuzzy relationship matrix

Establishing a membership function is the basis of fuzzy comprehensive evaluation. Considering the simplicity of practical operation and combining with the engineering background of groundwater quality evaluation, the “reduced half trapezoidal stepwise method” is generally adopted [21]. According to the Chinese national groundwater quality standard (GB/T 14848-2017), groundwater is divided into five levels. The formula for the grade of membership of water quality is as follows:

Class I:(1)$$\begin{array}{l} r_{i1}=\left\{ \begin{array}{ll} 1 & x_{i}\leq s_{i1}\\ \frac{s_{i2}-x_{i}}{s_{i2}-s_{i1}} & s_{i1}<x_{i}<s_{i2}\\ 0 & x_{i}>s_{i1} \end{array}\right. \end{array}$$
For class II~IV:(2)$$r_{ij}=\left\{ \begin{array}{ll} 1-\frac{s_{ij}-x_{i}}{s_{ij}-s_{ij-1}} & s_{ij-1}\leq x_{i}\leq s_{ij}\\ 0 & x_{i}\leq s_{ij-1},x_{i}>s_{ij+1}\\ \frac{s_{ij+1}-x_{i}}{s_{ij+1}-s_{ij}} & s_{ij}<x_{i}<s_{ij+1} \end{array}\right.$$
Class V:(3)$$r_{ij}=\left\{ \begin{array}{ll} 0 & x_{i}\leq s_{i4}\\ 1-\frac{s_{i5}-x_{i}}{s_{i5}-s_{i4}} & s_{in-1}<x_{i}<s_{in}\\ 1 & x_{i}>s_{i5} \end{array}\right.$$

Note: *x_i_* is the measured concentration of the *i*-th evaluation index; *s_ij_* is the standard value of grade *j* water quality of the *i*-th evaluation index; *r_ij_* is the membership of the *i*-th evaluation index to *j*-grade water quality.

The fuzzy relation evaluation matrix *R* can be determined from the membership function established above, namely:(4)$$R=\left(\begin{array}{ccc} r_{11} & \ldots & r_{15}\\ \vdots & \ddots & \vdots \\ r_{51} & \ldots & r_{55} \end{array}\right)$$

(3) Determine the weight coefficient matrix

Since there are many factors affecting water quality, and the influence of various factors on water quality is different, it is necessary to calculate the weight of each factor to make the evaluation model more scientific [22]. In this paper, the weight coefficient is determined by the entropy weight method.

Standardize raw data. The original data consists of *n* evaluation indicators, and *m* evaluation objects form an *X* matrix:$$X=\left(\begin{array}{llll} x_{11} & x_{12} & \ldots & x_{1m}\\ \vdots & \vdots & \vdots \\ x_{n1} & x_{n2} & \ldots & x_{nm} \end{array}\right)$$
Use the formula:(5)$$y_{ij}=\frac{\underset{j}{\max}\left\{x_{ij}\right\}-x_{ij}}{\underset{j}{\max}\left\{x_{ij}\right\}-\underset{j}{\min}\left\{x_{ij}\right\}}$$
Standardize and get the judgment matrix *Y*:$$Y=\left(\begin{array}{llll} y_{11} & y_{12} & \ldots & y_{1m}\\ \vdots & \vdots & \vdots \\ y_{n1} & y_{n2} & \ldots & y_{nm} \end{array}\right)$$
Entropy weight calculation formula:(6)$$w_{ei}=\frac{1-H_{i}}{n-\sum_{i=1}^{n}H_{i}}$$
where:$$H_{i}=\frac{-\sum_{j=1}^{m}f_{ij}\ln f_{ij}}{\ln m}$$
$$f_{ij}=(1+y_{ij})/\sum_{j=1}^{m}\left(1+y_{ij}\right)$$

(4) Establish fuzzy comprehensive evaluation model

The fuzzy comprehensive evaluation is a composite calculation of the fuzzy relation matrix and the weight coefficient matrix, namely:(7)B=W×R

The fuzzy matrix *B* is the membership degree of each water sample to different levels of water quality, of which the highest grade is the water quality level of the sample.

## 4. Results and Discussion

### 4.1. Analysis of Water Chemical Content Characteristics

Statistical analysis of pH, TDS, Na^+^, K^+^, Ca^2+^, Mg^2+^, SO_4_^2^^−^, Cl^−^, and HCO_3_^−^ in the Taiyuan formation limestone water of the Sulin mining area in the Huaibei coalfield and characteristic values are shown in Table 2.

It can be seen that the pH range of the water samples in the study area is 6.78~8.24, and all samples are in line with the limits of pH in the World Health Organization (WHO) guideline value (2011) and the Chinese national standard (GB5749-2006). The TDS values ranged from 327 to 686 mg/L. All samples met the Chinese national standard (GB5749-2006), but 81.25% of the water samples exceeded the WHO guidance value (2011). Ca^2+^, SO_4_^2^^−^, HCO_3_^−^, and Cl^−^ all have large coefficients of variation, indicating that they are subject to strong external disturbances, and the impact of coal mining is the first to bear the brunt.

The measured values of main cations and anions in the water were plotted in Figure 2. Where the ordinate is logarithmic coordinates, indicating the ion content, the abscissa is the sample number, representing different coal mines.

As can be seen from Figure 2, the content of cation in the water in the study area changes to Na^+^ > Mg^2^^+^ > Ca^2^^+^ > K^+^, and the content of anion changes to SO_4_^2^^−^ > HCO_3_^−^ > Cl^−^. Na^+^ is the main cation in limestone water, and the content ranges from 302.71 to 1872.59 mg/L. The K^+^ content is relatively low, but 68.75% of the samples still exceed the WHO guidance value (2011), indicating that the limestone water has high salinity. The contents of Ca^2^^+^ and Mg^2^^+^ were 5.06~496.19 mg/L and 6.94~339.27 mg/L, respectively. The Ca^2^^+^ concentrations of seven samples and the Mg^2^^+^ concentrations of three samples met the demands of the WHO (2011). SO_4_^2^^−^ is the main anion in limestone water; the content is between 4.4~3736.16 mg/L, and the over-standard rate is 81%. The Cl^−^ content is 78.76~1285.72 mg/L, and the over-standard rate is 31%.

The higher concentration of SO_4_^2^^−^ and the higher rate of exceeding the standard may be related to the dissolution of gypsum and the deposition of other sulfate minerals [25]. In addition, the Taiyuan formation limestone in the study area contains thin coal seams, and the pyrite content in the coal is rich. The percolation of pyrite through groundwater after oxidation may be another reason for the high concentration of SO_4_^2^^−^ in the Taiyuan formation limestone water [4].

Based on the SO_4_^2−^ content test data in the Taiyuan formation limestone water of the study area, the SO_4_^2^^−^ contour was drawn using Surfer software (Figure 3). It can be seen that the content of SO_4_^2^^−^ decreased gradually from north to south, and the maximum value of SO_4_^2^^−^ appeared in the Tongting–Linhuan coal mine area of the Linhuan mining area, which was basically consistent with the early research results of Yang et al. [7,26]. It can be seen from Figure 2 and Figure 3 that HCO_3_^−^ in the Taiyuan formation limestone water of the Sunan mining area is higher than that of the Linhuan mining area, while SO_4_^2^^−^ in the Taiyuan formation limestone water of the Linhuan mining area is higher than that of the Sunan mining area. According to the analysis, this is related to the geological tectonic setting and mining impact. When the Taiyuan formation limestone water is in different tectonic settings, its openness is different. There are also differences in the “desulfurization” (SO_4_^2^^−^ + 2C + 2H_2_O = H_2_S + 2HCO_3_^−^) [27]. The mining intensity of the coal mine at the bottom of the Permian system in the Sunan mining area is relatively large, and the disturbance degree to the flow field of the Taiyuan formation limestone water is also relatively large. The alternating cycle of the Taiyuan formation limestone water is accelerated, and the carbon source is rich, which promotes the process of desulfurization, and makes the content of HCO_3_^−^ and SO_4_^2^^−^ in the Taiyuan formation limestone water relatively high. The Linhuan mining area mainly mines the middle and upper coal seams of the Permian, and the development scale of the coal seams at the bottom of the Permian is small. Therefore, the disturbance of the flow field of the Taiyuan formation limestone water is small, the aquifer is relatively closed, the groundwater flow rate is slow, and the carbon source could not be replenished in time, and the effect of “desulfurization” was weakened, so the phenomenon of high SO_4_^2^^−^ and low HCO_3_^−^ in the Linhuan mining area was formed.

### 4.2. Mechanism Analysis of Water and Rock Action

The Piper trigram is a method to reveal the chemical characteristics of water and its evolution process [28]. The hydrogeochemical division and meaning represented by the rhombic area are shown in Figure 4. In order to understand the chemical evolution mechanism of the Taiyuan formation limestone water in the Sulin mining area of the Huaibei coalfield, the Piper trigram was drawn by Aquachem 4.0, as shown in Figure 5.

As can be seen from Figure 5, water sample points are mostly concentrated in zone 7 (see Figure 4). The content of alkali metal ions is higher than that of alkaline earth metal ions, and the content of strong acid roots is higher than that of weak acid roots. For cations in the Taiyuan formation limestone water, the distribution of water sample points in the Sulin mining area is relatively concentrated. In Figure 5, 75% of the water sample points fall in area D where Na^+^ is dominant. The remaining 25% of the water sample falls in zone B, which is mixed water. For anions, 62% of the water spots fall in zone F where SO_4_^2^^−^ dominates. Additionally, 19% of the water sample falls in zone E where HCO_3_^−^ dominates and 6% falls in zone G where Cl^−^ dominates. The remaining 12% of the water samples fall in zone B and belong to mixed water. It is concluded that sulfate mineral dissolution is the most important factor in controlling the chemical characteristics of the Taiyuan formation limestone water. Dissolution of carbonate minerals and evaporative salts also plays a role. In the water samples from the Linhuan mining area, 78% of the water samples were SO_4_·Cl-Na type, and all the water samples from the Sunan mining area were HCO_3_-Na type.

Hydrochemical formation of the Taiyuan formation limestone water in the study area can be understood by using Gibbs semi-logarithmic coordinates (Figure 6). Its ordinate is TDS; the abscissa is Na^+^/(Na^+^ + Ca^2+^) or Cl^−^/(Cl^−^ + HCO_3_^−^).

It can be seen from Figure 6 that the TDS value of the Taiyuan formation limestone water is between 100 and 1000 mg/L, and the Na^+^/(Na^+^ + Ca^2^^+^) range is 0.45 to 0.99; that is, the content of Na^+^ in the water is higher than that of Ca^2^^+^. The range of Cl^−^/(Cl^−^ + HCO_3_^−^) is 0.09~0.91, most of which is concentrated around 0.5; that is, the content of Cl^−^ and HCO_3_^−^ in water is basically equal. Moreover, most of the water samples are concentrated in the rock weathering control area, so it can be considered that rock weathering is the main factor affecting the hydrochemical characteristics of the Taiyuan formation limestone water in the study area. Some of the water samples in Figure 6 (especially in Figure 6a) fall outside the solid line, indicating that there are other effects controlling the chemical composition of groundwater, such as cation exchange [29].

No water sample points fall in the precipitation area, indicating that there is no direct hydraulic connection between the Taiyuan formation limestone water and atmospheric precipitation in the study area. Chen et al. [30] conducted hydrogen and oxygen stable isotope tests on 30 groundwater samples from different aquifers in the Linhuan mining area, and the results showed that there was generally no direct hydraulic connection between the Taiyuan formation limestone water and other aquifers. Therefore, the analysis results of this Gibbs diagram are consistent with the research results of Chen et al.

Common forms of water–rock interaction are silicate weathering, evaporating salt, and carbonate dissolution. In general, the main types of water–rock interaction can be determined by the correlation ratios of Ca^2+^, Mg^2+^, Na^+^, and HCO_3_^−^ [31]. It can be seen from Figure 7 that the water samples in the study area are mainly distributed between evaporite dissolution and silicate weathering, indicating that the water samples in the study area are related to evaporite dissolution and silicate weathering, but the contribution of carbonate dissolution to the composition of water samples is not excluded. The distribution of water samples in the Sunan mining area is relatively scattered, which indicates that groundwater in different coal mines is affected by evaporite dissolution and silicate weathering to different degrees.

The molar ratio of major ions is widely used in the identification of groundwater hydrogeochemical processes and formation mechanisms [32]. The dissolution of halite releases roughly equal amounts of Na^+^ and Cl^−^ in water, so the ratio of Na^+^ to Cl^−^ is often used to determine the dissolution of halite in water [33]. The molar ratio of the main ions of the Taiyuan formation limestone water in the Sulin mining area of the Huaibei coalfield is shown in Figure 8. In the Na^+^ Cl^−^ figure (Figure 8a), all water samples are above the 1:1 trend line, that is, Na^+^ content in the Taiyuan formation limestone water is higher than Cl^−^, indicating that halite dissolution is not the only source of Na^+^ [34]. Na^+^ has other sources, such as the dissolution of minerals [35]. In the scale coefficient diagram of Ca^2+^ + Mg^2+^—HCO_3_^−^ (Figure 8b), most of the water samples are distributed above the 1:1 trend line, that is, the Ca^2+^ + Mg^2+^ content is higher than HCO_3_^−^, indicating that the carbonate dissolution is not the main source of Ca^2+^ and Mg^2+^. Figure 8c shows that Ca^2+^ has a good correlation with Mg^2+^, and Mg^2+^ content is larger than Ca^2+^, indicating that not only is dolomite dissolved, but also the dissolution of magnesite may exist. In Figure 8d, all water sample points are distributed below the 1:1 trend line, that is, the content of SO_4_^2^^−^ is higher than that of Ca^2+^. According to the analysis of the proportional coefficient diagram of Ca^2+^ + Mg^2+^—SO_4_^2^^−^ (Figure 8e), 63% of the water sample points are near the 1:1 trend line, indicating that Ca^2+^ and Mg^2+^ mainly come from sulfate dissolution, but a few of the water sample points show that the content of SO_4_^2^^−^ is greater than that of Ca^2+^ + Mg^2+^, indicating that SO_4_^2−^ may have other sources, such as the oxidation of pyrite. The results of statistical analysis show that when the ratio of (Ca^2+^ + Mg^2+^)/(HCO_3_^−^ + SO_4_^2^^−^) was much less than 1, the ionic components in groundwater mainly came from the dissolution of sulfate. When the ratio of (Ca^2+^ + Mg^2+^)/(HCO_3_^−^ + SO_4_^2^^−^) is much larger than 1, the ionic component in groundwater is mainly derived from the dissolution of carbonate. When the ratio of (Ca^2+^ + Mg^2+^)/(HCO_3_^−^ + SO_4_^2^^−^) is close to 1, the ionic component in groundwater is affected by both carbonate dissolution and sulfate dissolution [36]. As can be seen from Figure 8f, the (Ca^2+^ + Mg^2+^)/(HCO_3_^−^ + SO_4_^2^^−^) ratio is much smaller than 1, so the ionic component in the Taiyuan formation limestone water is mainly derived from the dissolution of sulfate.

### 4.3. Fuzzy Comprehensive Evaluation of Water Quality

In this paper, the improved fuzzy comprehensive evaluation method is adopted, and the groundwater quality standard (GB/T 14848-2017) is used as the evaluation standard. The four parameters of TDS, Na^+^, Cl^−^, and SO_4_^2^^−^ are selected to check the water quality of the Taiyuan formation limestone water in the Sulin mining area.

Taking the S13 test data as an example, the membership degree can be calculated according to Formulas (1)–(3) to determine the fuzzy relation matrix:$$R=\left(\begin{array}{ccccc} 0 & 0.852 & 0.148 & 0 & 0\\ 0 & 0 & 0.286 & 0.714 & 0\\ 0 & 0.132 & 0.868 & 0 & 0\\ 0 & 0 & 0 & 0 & 1 \end{array}\right)\begin{array}{c} \mathrm{TDS}\\ Na^{+}\\ {\mathrm{Cl}}^{-}\\ {{\mathrm{SO}}_{4}}^{2-} \end{array}$$

According to the entropy weight method, Formula (6) is used to calculate the weight of different evaluation factors in the fuzzy comprehensive evaluation:*W* = (0.3625, 0.1721, 0.1504, 0.3149)

According to Formula (7), the membership degree of S13 to various quality levels of groundwater is calculated:*B* = *W* × *R* = (0, 0.3287, 0.2366, 0.1229, 0.3149)

According to the principle of maximum membership, 0.3287 is the maximum of the five numbers. So S13 belongs to class II.

According to the above steps, the fuzzy comprehensive evaluation of other water samples was carried out, and the calculation results are shown in Table 3.

As can be seen from Figure 9, I, II, and III types of water account for 12.5%, respectively, which is suitable for industrial, agricultural, and drinking water. The remaining 62.5% of the water samples for V class water was of poor quality and required treatment before it could be used for industrial, agricultural, and drinking water supply [20].

In the study area, the proportion of V type water is relatively large, which is related to Na^+^ and SO_4_^2^^−^ exceeding the standard, in which the rate of Na^+^ exceeding the standard is 68.75% and the rate of SO_4_^2^^−^ exceeding the standard is 75% (SO_4_^2^^−^ maximum exceeding the standard multiple is 10). Na^+^ mainly came from the dissolution of halite and cation exchange, SO_4_^2^^−^ mainly came from the dissolution of sulfate and the oxidation of pyrite. The proportion of V water in the Taiyuan formation limestone water in the Sunan and Linhuan mining areas was 50% and 70%, respectively, indicating that the water quality of the Sunan mining area was better. The reason is that the carboniferous Taiyuan formation aquifer in the Sunan mining area is relatively open and has sufficient carbon source supplement. The “desulfurization” effect was strengthened, resulting in a decrease in the SO_4_^2^^−^ content in the Taiyuan formation limestone water.

## 5. Conclusions

(1) The content of cations in the Taiyuan formation limestone water of the study area changed to Na^+^ > Mg^2+^ > Ca^2+^ > K^+^, and the content of anions changed to SO_4_^2^^−^ > HCO_3_^−^ > Cl^−^. Under the influence of different geological structure backgrounds and mining disturbances, the “desulfurization” effect of the Taiyuan formation limestone water in the Sulin mine area is different. As a result, the SO_4_^2^^−^ in the Taiyuan formation limestone water of the Linhuan mining area is higher than the Sunan mining area, while the HCO_3_^−^ in the Sunan mining area is higher than the Linhuan mining area.

(2) Based on the Piper trigram, the hydrochemical types of water in the study area are mainly SO_4_·Cl-Na, HCO_3_-Na, and SO_4_·Cl-Ca·Mg. The main types of water in the Linhuan and Sunan mine are SO_4_·Cl-Na and HCO_3_-Na, respectively.

(3) The results of Gibbs diagram analysis show that weathering and leaching are the main factors affecting the hydrochemical characteristics of the water in the study area. Based on the ratios of the main ions, it was determined that evaporation salt dissolution and silicate weathering were the main water–rock processes. The dissolution of rock salt is not the only source of Na^+^, Ca^2+^ and Mg^2+^ mainly come from the dissolution of sulfate, but the dissolution of sulfate is not the only source of SO_4_^2^^−^.

(4) The results of fuzzy comprehensive evaluation showed that Class I, Class II, and Class III water in water of the study area accounted for 12.5%, respectively, and the remaining 62.5% of the water samples were Class V. The Taiyuan formation limestone water in the Sunan mining area is slightly better than the Linhuan mining area. The evaluation results show that the ratio of V type water is high, and the water quality is poor, not suitable for direct use, and will cause different types of harm to different purposes; it must be treated before the water can be used.

## Figures and Tables

**Figure 1 ijerph-16-02512-f001:**
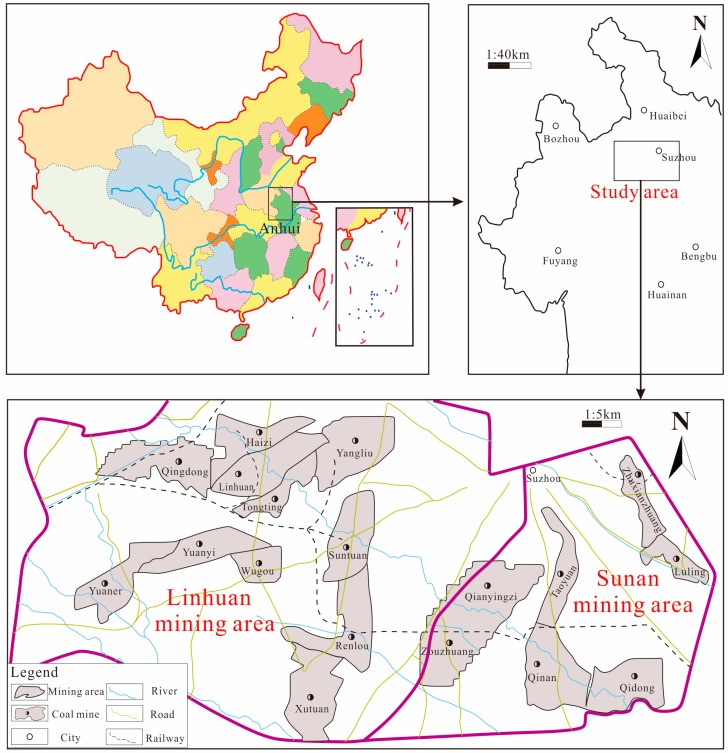
Geographical location of the Sulin mining area.

**Figure 2 ijerph-16-02512-f002:**
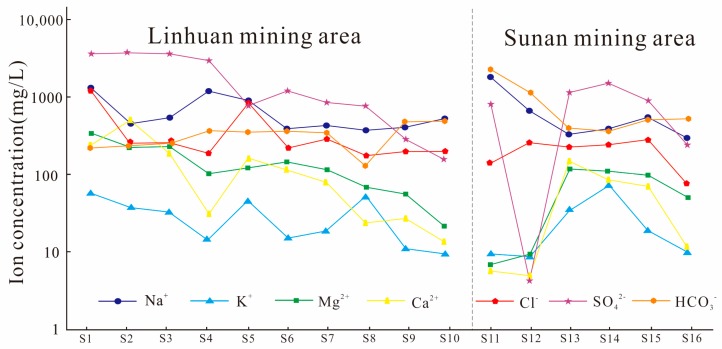
Variation characteristics of various indicators of the Taiyuan formation limestone water in the study area.

**Figure 3 ijerph-16-02512-f003:**
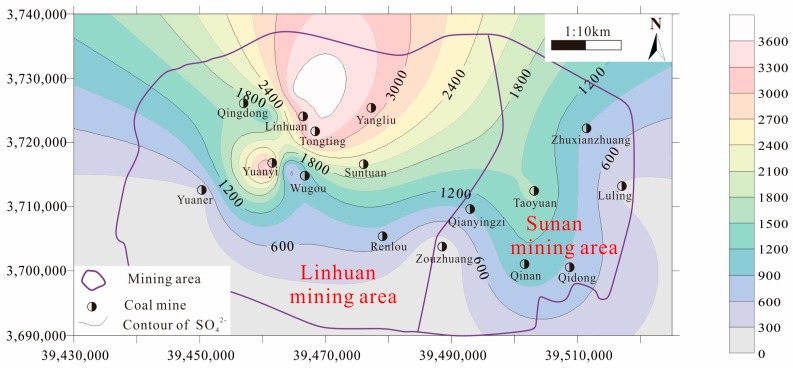
SO_4_^2^^−^ contour map of the Taiyuan formation limestone water in the study area.

**Figure 4 ijerph-16-02512-f004:**
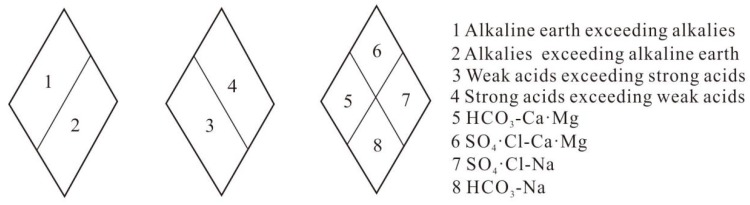
Piper trigram diamond-shaped hydrogeochemical map.

**Figure 5 ijerph-16-02512-f005:**
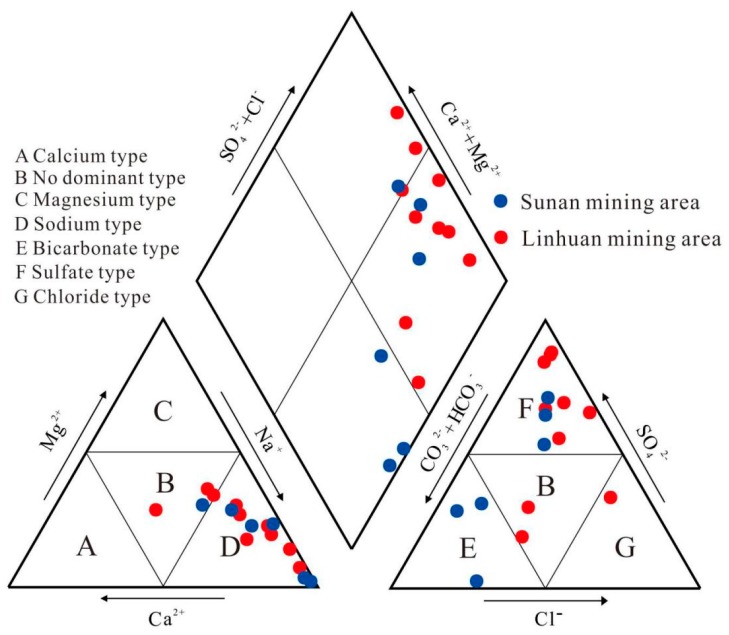
Piper trigram of the Taiyuan formation limestone water constant components in the study area.

**Figure 6 ijerph-16-02512-f006:**
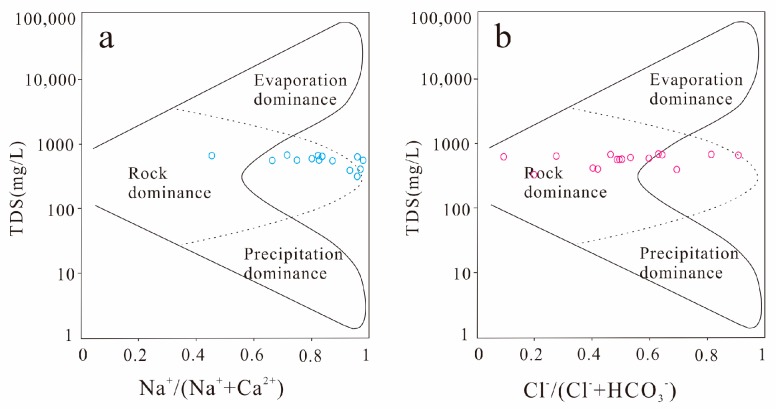
Gibbs diagram of the Taiyuan formation limestone water in the study area.

**Figure 7 ijerph-16-02512-f007:**
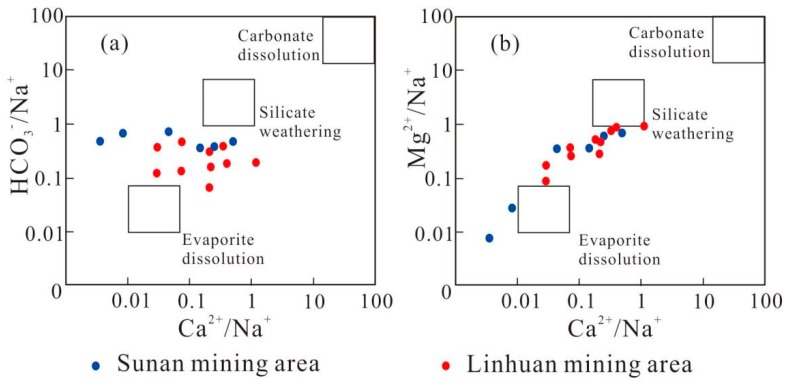
Diagram of ion combination ratio of the Taiyuan formation limestone water in the study area.

**Figure 8 ijerph-16-02512-f008:**
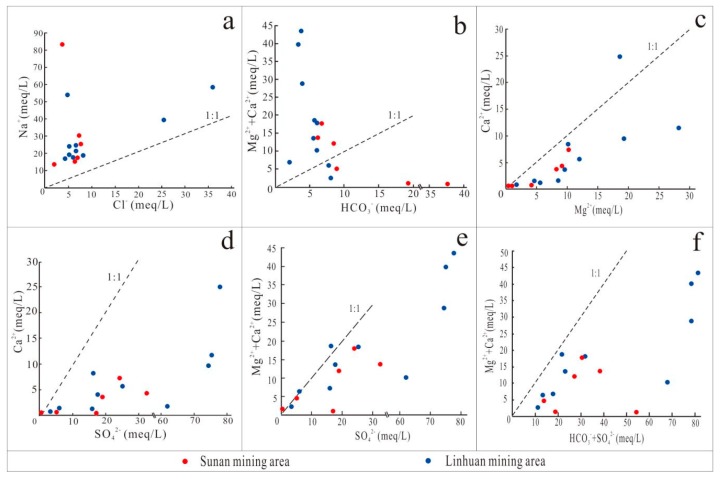
Molar ratio diagram of the Taiyuan formation limestone water ions in the study area. (Na^+^ and Cl^−^ (**a**), Mg^2+^ + Ca^2+^ and HCO_3_^−^ (**b**), Ca^2+^ and Mg^2+^ (**c**), Ca^2+^ and SO_4_^2−^ (**d**), Mg^2+^ + Ca^2+^ and SO_4_^2^^−^ (**e**), Mg^2+^ + Ca^2+^ and HCO_3_^−^ + SO_4_^2^^−^ (**f**)).

**Figure 9 ijerph-16-02512-f009:**
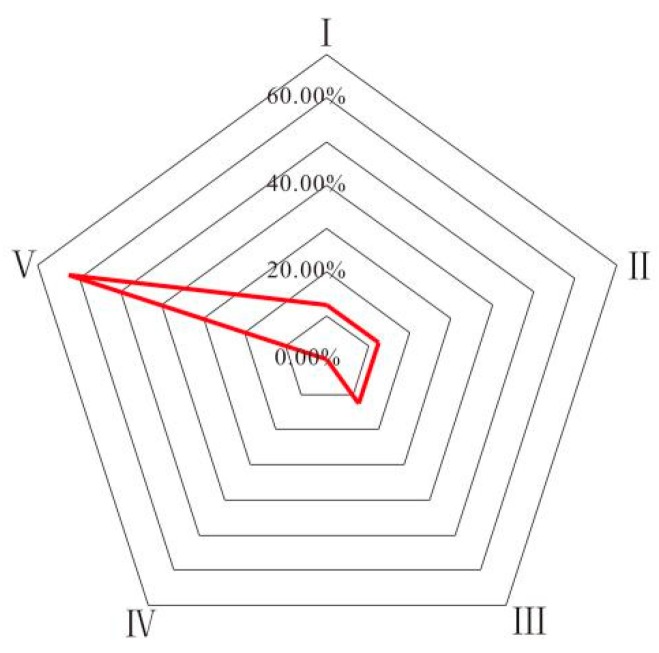
Radar chart of fuzzy comprehensive evaluation results.

**Table 1 ijerph-16-02512-t001:** Groundwater quality classification based on the Chinese national standard (GB/T 14848-2017). TDS: total dissolved solids.

Grade	Classification	Parameters (Unit: mg/L)			
		TDS	Na^+^	Cl^−^	SO_4_^2^^−^
I	Excellent suitable for drinking water	≤300	≤100	≤50	≤50
II	Good suitable for drinking water	≤500	≤150	≤150	≤150
III	Moderate suitable for drinking water	≤1000	≤200	≤250	≤250
IV	Poor suitable for drinking water	≤2000	≤400	≤350	≤350
V	Unsuitable suitable for drinking water	>2000	>400	>350	>350

**Table 2 ijerph-16-02512-t002:** Statistical characteristics of constant composition of the Taiyuan formation limestone water.C.V:Coefficient of variation.

Paramaters	Units	Min	Max	Mean	C.V(%)	a	b
pH	–	6.78	8.24	7.31	5.3	6.5–8.5	6.5–8.5
TDS	mg/L	327	686	567.63	20.04	500	1000
Na^+^	mg/L	302.72	1872.59	667.20	47.55	200	200
K^+^	mg/L	8.92	75.69	28.44	70.46	12	−
Mg^2+^	mg/L	6.94	339.27	114.27	75.33	50	−
Ca^2+^	mg/L	5.06	496.19	106.44	114.69	70	−
Cl^−^	mg/L	78.76	1285.72	326.64	93.02	250	250
SO_4_^2^^−^	mg/L	4.40	3736.16	1416.64	89.06	250	250
HCO3^−^	mg/L	129.97	2297.29	534.85	94.92	500	−

Note: a: World Health Organization (WHO) guideline value (2011) [23]; b: national standard (GB5749-2006) [24].

**Table 3 ijerph-16-02512-t003:** Fuzzy comprehensive evaluation results of the Taiyuan formation limestone water quality in the study area.

Sample	I	II	III	IV	V	Water Quality Level
S1	0	0.2422	0.1204	0	**0.6375**	V
S2	0	0.2411	0.2719	0	**0.4870**	V
S3	0	0.2277	0.2853	0	**0.4870**	V
S4	0	0.3276	0.1854	0	**0.4870**	V
S5	0	0.2306	0.1320	0	**0.6375**	V
S6	0	0.0376	**0.4341**	0.2134	0.3149	III
S7	0	0.0812	0.3701	0.0616	**0.4870**	V
S8	0	0.3032	0.2360	0.1532	**0.3149**	V
S9	0.1849	0.2529	**0.2852**	0.1049	0.1721	III
S10	0.1396	**0.5878**	0.1005	0	0.1721	II
S11	0.0120	0.4132	0.0877	0	**0.4870**	V
S12	**0.3149**	0.2661	0.2274	0.0194	0.1721	I
S13	0	**0.3287**	0.2336	0.1229	0.3149	II
S14	0.0120	0.2893	0.0847	0.3111	**0.3149**	V
S15	0	0.3154	0.1449	0.0527	**0.4870**	V
S16	**0.4208**	0.0922	0.0996	0.3874	0.0000	I

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
