# Peer review of "Hydrogeochemical Characteristics and Water Quality Evaluation of Carboniferous Taiyuan Formation Limestone Water in Sulin Mining Area in Northern Anhui, China"

_ijerph, 2019, doi:10.3390/ijerph16142512_

Round 1

Reviewer 1 Report

A very clear description of methodology and results. 

The study shows implications for the industrial, agriculture and drinking water quality. It would be useful if you could provide some literature view about the potential impacts of poor quality on each of these sectors and then suggest some policy recommendations to improve the conditions for each sector (just another short paragraph could be added in conclusion). 

Author Response

Response to Reviewer 1 Comments

Point 1: The study shows implications for the industrial, agriculture and drinking water quality. It would be useful if you could provide some literature view about the potential impacts of poor quality on each of these sectors and then suggest some policy recommendations to improve the conditions for each sector (just another short paragraph could be added in conclusion). 

Response 1: OK.Thank you for your suggestion. I have  made some minor changes to conclusion 4 and added some content about the potential impacts of poor quality on each of these sectors and how to improve the conditions for each sector.Thank you again.

Reviewer 2 Report

In this paper, the authors adopted a set of traditional methods including Piper trigram, Gibbs diagram, ion scale coefficient map and fuzzy comprehensive manner. The authors completed this paper well-structured. However, there are still some problems to be further improved and discussed as well:

1. The content of the article is more like a report of production which following a particular process rather than a scientific article. Also, little novel findings can be found in this manuscript.

2. The authors need to have their work reviewed by a proper translation/reviewing service before submission.

3. For the Abstract part, the content about implication/possible application/limitation or future work should be added.

4. For Fig.1, the whole picture is not readable enough duo to the covering lines in every color. It is suggested to use some color blocks to separate the different region in the study area.

5. For part 3.1 sampling and testing, there’s no information about the sampled water, is it the surface water or groundwater? If it’s groundwater, confined water or phreatic water? Also, the name for the sampled water is so complicated, number code would be better. Besides, further information which including name, model and accuracy about the portable instrument is needed.

6. For Fig.2, first, it’s not easy to distinguish the lines just by color, some different symbols on the line are needed. The most important problem about the picture is insufficient interpretation of the data, which means readers can’t get any useful information and results directly and immediately.

7. For Fig.3, you should think about whether the number of water samples in this article is sufficient to draw a map based on interpolation method. There is a high concentration area in the northern part of the model, but there is no control point.

8. Formula (8) is missing from the text.

9. Line 358-359, “Only through treatment can it be used as industrial, agricultural and drinking water supply”. The conclusion of this conclusion is not rigorous enough, because different standards of groundwater have different uses, they cannot be generalized.

Author Response

Response to Reviewer 2 Comments

Point 1: The content of the article is more like a report of production which following a particular process rather than a scientific article. Also, little novel findings can be found in this manuscript.

Response 1: Maybe there is no outstanding innovation in this paper, but I focus on the mining area and take a special kind of groundwater (the Taiyuan formation limestone water) as the research object, which I think is innovation.The formation mechanism and water quality of the Taiyuan formation limestone water are studied in order to provide reference for the protection and rational exploitation.

Point 2: The authors need to have their work reviewed by a proper translation/reviewing service before submission.

Response 2: Thanks for the warning. I have invited my English teacher revise my paper in order to make my articles easier for readers to understand.

Point 3: For the Abstract part, the content about implication/possible application/limitation or future work should be added.

Response 3: OK.I have already added the importance of this research in the end of Abstract.

Point 4: For Fig.1, the whole picture is not readable enough duo to the covering lines in every color. It is suggested to use some color blocks to separate the different region in the study area.

Response 4: Thank you for your advice.Now I have used some color blocks to separate the different region in the study area.

Point 5: For part 3.1 sampling and testing, there’s no information about the sampled water, is it the surface water or groundwater? If it’s groundwater, confined water or phreatic water? Also, the name for the sampled water is so complicated, number code would be better. Besides, further information which including name, model and accuracy about the portable instrument is needed.

Response 5: Maybe I need to explain,the Taiyuan formation limestone water belongs to groundwater.And it is also confined water.I have used number code to correct the name for the sampled water.I have added information which including name, model and accuracy about the portable instrument.

Point 6:For Fig.2, first, it’s not easy to distinguish the lines just by color, some different symbols on the line are needed. The most important problem about the picture is insufficient interpretation of the data, which means readers can’t get any useful information and results directly and immediately.

Response 6: According to your suggestion,I change the sign on each line so that readers can easily distinguish different lines. At the same time,I give some new interpretation of the data to make readers can get some useful information directly.

Point 7: For Fig.3, you should think about whether the number of water samples in this article is sufficient to draw a map based on interpolation method. There is a high concentration area in the northern part of the model, but there is no control point.

Response 7:I think the number of water samples in this paper is sufficient to draw a map based on interpolation method. I think I need to explain to you why there is a high concentration area, but there is no control point. Because my research object is the Taiyuan formation limestone water, a groundwater buried several hundred meters underground. So I can only sample at fixed points, I can't sample anywhere I want. For example, the point you mentioned.And we also can find the SO42- concentration of deep groundwater from the coal-bearing aquifer of the Linhuan is high by reading article “Hydrochemical characteristics and quality assessment of deep groundwater from the coal-bearing aquifer of the Linhuan coal-mining district, Northern Anhui Province, China”. Although our research objects are not exactly the same, there is a hydraulic connection between the two kinds of deep groundwater, so I think there is no problem to believe in Fig.3.

Point 8:Formula (8) is missing from the text.

Response 8: I am Sorry. When I wrote the article, I marked the wrong formula number.It has been corrected.

Point 9:Line 358-359, “Only through treatment can it be used as industrial, agricultural and drinking water supply”. The conclusion of this conclusion is not rigorous enough, because different standards of groundwater have different uses, they cannot be generalized.

Response 9: When I wrote this sentence, my mean it has been treated differently for different purposes. Maybe I didn't express myself clearly enough to make you understand.Now I have deleted this sentence.Now I have changed the sentence to make the meaning more clear.